# In-Season Quantification and Relationship of External and Internal Intensity, Sleep Quality, and Psychological or Physical Stressors of Semi-Professional Soccer Players

**DOI:** 10.3390/biology11030467

**Published:** 2022-03-18

**Authors:** Hadi Nobari, Roghayyeh Gholizadeh, Alexandre Duarte Martins, Georgian Badicu, Rafael Oliveira

**Affiliations:** 1Department of Physiology, Faculty of Sport Sciences, University of Extremadura, 10003 Cáceres, Spain; 2Sports Scientist, Sepahan Football Club, Isfahan 81887-78473, Iran; 3Department of Exercise Physiology, Faculty of Educational Sciences and Psychology, University of Mohaghegh Ardabili, Ardabil 56199-11367, Iran; r.gholizadeh@uma.ac.ir; 4Sports Science School of Rio Maior—Polytechnic Institute of Santarém, 2040-413 Rio Maior, Portugal; alexandremartins@esdrm.ipsantarem.pt (A.D.M.); rafaeloliveira@esdrm.ipsantarem.pt (R.O.); 5Life Quality Research Centre, 2040-413 Rio Maior, Portugal; 6Comprehensive Health Research Centre (CHRC), Departamento de Desporto e Saúde, Escola de Saúde e Desenvolvimento Humano, Universidade de Évora, Largo dos Colegiais, 7000 Évora, Portugal; 7Department of Physical Education and Special Motricity, Faculty of Physical Education and Mountain Sports, Transilvania University of Braşov, 500068 Braşov, Romania; georgian.badicu@unitbv.ro; 8Research Centre in Sport Sciences, Health Sciences and Human Development, 5001-801 Vila Real, Portugal

**Keywords:** load, heart rate, high-speed running, monotony, muscle soreness, sprint, sleep, strain, stress

## Abstract

**Simple Summary:**

In this study, we analyse the relationship of the in-season variations of external, internal and well-being measures across different periods of a semi-professional soccer season (early-, mid- and end-season) and describe TM and TS for the entire period of the analysis. The main findings of our study revealed that increasing the training intensity affects the well-being of the players and consequently the training intensity management. Coaches and their staff should consider the results of this study, because despite the relationship between external and internal intensity, each has a unique effect on the perception of the player’s training intensity management.

**Abstract:**

The purpose of this study was two-fold: (a) to describe and analyse the relationship of the in-season variations of external and internal intensity metrics as well as well-being measures across different periods of a semi-professional soccer season (early-, mid- and end-season); and (b) to describe training monotony (TM) and training strain (TS) for 20 weeks in a semi-professional soccer season. Eighteen semi-professional players (age: 29 ± 4.1) from the Asian First League team participated in this study. The players were monitored for 20 consecutive weeks during in-season for external training intensity, internal training intensity and well-being parameters. The in-season was organized into three periods: early-season (weeks 1–7); mid-season (weeks 8–13); and end-season (weeks 14–20). Total distance (TD), high-speed running distance (HSRD), sprint distance, rate of perceived exertion (RPE), session-RPE (s-RPE), TM, TS, heart rate average and maximum, as well as sleep quality, stress and muscle soreness were collected. Results revealed that TD, HSRD and sprint distance (total values) were meaningfully greater during end-season than in the early-season. RPE showed a significantly highest value during the end-season (4.27 AU) than in early- (3.68 AU) and mid-season (3.65 AU), *p* < 0.01. TS showed significant differences between early-season with mid-season (*p* = 0.011) and end-season (*p* < 0.01), and the highest value occurred in week 17 during end-season (6656.51 AU), while the lowest value occurred in week 4 during early-season (797.17 AU). The average TD periods showed a moderate to large correlation with RPE, sleep and s-RPE at early-, mid- and end-season. Increasing the training intensity without considering the well-being of the players affects the performance of the team. Examining processes of the relationship between training intensity and other psychological indicators among players will probably be effective in training planning. Sports coaches and fitness professionals should be wary of changes in TM and TS that affect players performance. Therefore, to better control the training, more consideration should be given by the coaches.

## 1. Introduction

Nowadays, it is almost mandatory to monitor intensity and well-being to know the effects of exercise training programs on soccer players, and to individualize the training process in semi-professional soccer teams [1]. Through the season, there are several variations on training and well-being measures, and their monitoring is essential to apply the most successful strategy for recovery and competition [2].

In a recent study by Impellizzeri et al. [3], intensity monitoring definition was updated and defined in two dimensions: external (the physical demands imposed by the design and mode of exercise), and internal (the psychophysiological impact of external intensity). Usually, distances of different speed thresholds and accelerometry-based variables are the main measures reported as external intensity, while heart rate and rated perceived exertion (RPE) are the main measures reported as internal intensity [4].

Furthermore, well-being monitoring also represents a non-invasive valid and quick tool for collecting information associated with the status and readiness of the players to the training process and competition [5,6]. One example of an instrument that allows this monitoring process is the Hooper questionnaire [5], which include four categories: delayed onset muscle soreness (DOMS), fatigue, sleep, and stress.

Monitoring external, internal and well-being measures is part of daily strategies used to quantify the training session effects [6], but it can also allow the identification of intensity variations across the season [7]. In this sense, additional analyses can be performed to specific data obtained. For instance, two traditional indexes known as training monotony (TM) and training strain (TS) have been used to analyse such week variations. TM is used to analyse the intensity variability within the week, while TS is used to analyse the intensity variability multiplied by the accumulated intensity of the week [8]. The relationship between both indexes is supported by their formulas where TM is calculated by dividing the daily mean load by the standard deviation while TS is calculated through the product of weekly load by TM [8]. Usually, both indexes are based on the training duration multiplied by RPE (s-RPE) [8].

However, despite such a diversity of intensity measures, few studies have analysed the relationship between external, internal and well-being measures [9,10,11]. For instance, Haddad et al. [10] did not observe any relationship between the Hooper Index (HI) categories and RPE through submaximal exercises in junior soccer players, but Clemente et al. [9] showed negative correlations between s-RPE with DOMS, sleep, fatigue, and stress in weeks with two matches across training data from a full professional soccer season [10]. Moreover, Oliveira et al. [11] analysed associations in 10 in-season mesocycles (full-season) between all external, internal and wellbeing measures, and found negative correlations between stress and total distance [11]. In addition, the same authors found positive correlations between fatigue and s-RPE, between DOMS and s-RPE, and between HI total score and total distance [11].

The above-mentioned results were obtained in small sample sizes (n ranged between 17–35), which is common for soccer studies, and thus more research is needed to analyse the relationship between external, internal and well-being measures simultaneously, since there were multiple occasions where such analysis was not presented. Therefore, this study aims: (a) to describe and analyse the relationship of the in-season variations of external, internal and well-being measures across different periods of a semi-professional soccer season (early-, mid- and end-season); and (b) to describe TM and TS for the entire period of the analysis in a semi-professional soccer season.

## 2. Materials and Methods

### 2.1. Participants

In this study, eighteen male semi-professional soccer players from Iran’s First League were examined and monitored (age, 29 ± 4.1 years; height, 179.6 ± 4.7 cm; body mass, 74.9 ± 3.9 kg). All players participated in the in-season (>80%) [9]. 

The exclusion criteria were adopted from a previous study, namely, players with injury or that did not participate in training for at least two consecutive weeks and non-field positions such as the goalkeepers due to differences in intensity in training and matches [12]. All participants were familiarised with the training protocols prior to investigation. The Ardabil University of Medical Sciences’ ethical committee code IR.ARUMS.REC.1399.545 was authorized for this study, which followed the Declaration of Helsinki’s guidelines.

### 2.2. Experimental Design

A descriptive longitudinal approach of 20 in-season consecutive weeks was used. Specifically, for the present study, all players participated in 47 training sessions and 20 matches. Only data from regular training sessions was considered for analysis which means that data from resistance training, competitions, rehabilitation and/or recuperation sessions was excluded. All session were planned by the coach and staff, and the researchers only controlled the initial and final 30 min of the sessions. The analysed period ranged from early season (30 October 2017) until the end of the season (18 March 2018). The present in-season was organized into three periods: early-season (weeks 1–7); mid-season (weeks 8–13); and end-season (weeks 14–20) (Figure 1). Players’ weekly averages and accumulated values were used for analysis.

Figure 1 shows the distributions of weeks per periods of the season, as well as the number of training sessions and matches.

### 2.3. External Intensity Monitoring

All training sessions were monitored using GPS (GPSPORTS systems Pty Ltd., Model: SPI HPU; Canberra, Australia). The GPS included 15 Hz position GPS and a tri-axial accelerometer, and this device has previously shown high validity and reliability [13]. For data collection, belts were placed on the players’ shoulders and chests. At the end of the sessions, belts were collected and checked by the team’s GPS manager. Then, devices entered the dock system to download the information to the Team AMS software. After this procedure and before next session, all belts were recharged. The SPI IQ Absolutes were adjusted for GPS default zone throughout the season. 

The measures used for analysis were: TD; HSRD covered between 18 to 23 km/h^−1^; and sprint distance covered above >23 km/h^−1^. Data were considered in daily average values and the total of each period analysed, respectively.

### 2.4. Internal Intensity Monitoring

Players were monitored daily using a Borg’s CR10 scale [14], adapted by Foster et al. [15]. This scale showed validity and reliability for quantifying session intensity [16]. 

Thirty minutes after each session, players individually provided their RPE value using a tablet to avoid non-valid scores. The RPE values provided were also multiplied by the training duration, to obtain the s-RPE [15,17]. Previously, all players were familiarized with the RPE scale. 

Through s-RPE, TM (mean of training load during the seven days of the week divided by the SD of the training load of the seven days) [12,18,19] and TS (sum of the training load for all training sessions during a week multiplied by TM) [12,18,19] were calculated.

A flashing RED light was used to track HR. We placed each unit perpendicular to the bag, the logo on the unit was facing backwards and the RED light was on. HPUs are designed to automatically collect athlete’s HR data in one session. In addition to the GPS receiver, the SPI Pro X unit consists of a tri-axial accelerometer for estimating the forces on the player, and an integrated HR monitor. The following variable was selected: HR average (HRavg). Then, weekly HRavg was calculated by the average value for the entire week for each period, respectively. The way this information was recorded was similar to previous studies [12,20,21]. Daily average data were used for RPE, s-RPE and HRavg. 

### 2.5. Well-Being Monitoring

Approximately thirty minutes before sessions, players provided the HI scores [5] with the same procedures of the RPE. HI is a questionnaire that includes fatigue, stress, DOMS (scale of 1–7, in which 1 is very, very low and 7 is very, very high), and quality of sleep of the night that preceded the evaluation (scale of 1–7, in which 1 is very, very bad and 7 is very, very good). However, due to the purposes of the coach, fatigue was not considered in the present study. Daily average data was used for each category. 

### 2.6. Statistical Analysis

Descriptive statistics were used to characterize the sample. Shapiro–Wilk was used to test normality of results. Results were presented as mean ± SD. The relationship between all variables at the different periods was verified using bivariate correlations [22] (Pearson’s product–moment correlation coefficient (*r*)). The correlations’ effect size (ES) were calculated using the following criteria: <0.1, (trivial); 0.1–0.3, (small); >0.3–0.5, (moderate); >0.5–0.7, (large); >0.7–0.9, (very large); and >0.9, (virtually perfect) 

All variables obtained a normal distribution (Shapiro–Wilk, *p* > 0.05), a repeated measures ANOVA test was used with the Bonferroni post hoc test was used to compare variables for periods throughout the in-season. Statistical significance was set at *p* ≤ 0.05. Hedge’s g ES was also calculated to determine the magnitude of pairwise comparisons through the following formula: (mean 1–mean 2)/SD * pooled [23]. Then, the Hopkins threshold was applied: g ≤ 0.2, (trivial); 0.2 to ≤0.6, (small); 0.6 to ≤1.2, (moderate); 1.2 to ≤2.0, (large); 2.0 to ≤4.0, (very large); and >4.0, (nearly perfect) [24,25]. All data were analysed using IBM SPSS Statistics (version 22, IBM Corporation (SPSS Inc., Chicago, IL, USA).

## 3. Results

Table 1 shows the differences between the early-, mid- and end-season for all measures. 

To organize the results section, five sub-sections will address external, internal and well-being monitoring, correlations, monotony, and strain descriptions, respectively. To simplify the results description, only moderate to nearly perfect ES’s will be described here.

### 3.1. External Intensity Monitoring

In relation to TD (average values), a significant difference was found between early- vs. end-season, while TD in total values showed two significant differences, early- vs. end-season (*p* = 0.022), and mid- vs. end-season (*p* = 0.003), both with moderate ES. The HSRD (in average and total values) showed significant differences between all periods of the in-season. The sprint distance shows a significant difference between mid- vs. end-season.

### 3.2. Internal Intensity Monitoring

The RPE showed the highest values in the end-season (4.27 AU), with significant differences between the early- and mid-season. Average and total training duration showed the lower values in the early-season (58.91 and 989.61 min, respectively). They both showed significant differences with mid- and end-season. Regarding to s-RPE, it showed higher values in end-season (333.27 AU) compared to early- and mid-season. TS showed significant differences between early-season with mid- and end-season.

### 3.3. Well-Being Monitoring

There were no meaningful differences for quality of sleep, stress, or DOMS.

### 3.4. Correlations of All Measures for Each Period

Table 2, Table 3, Table 4 and Table 5 show the correlation coefficient of all measures in the study on the early-, mid-, end-season, and full-season, respectively.

### 3.5. Training Monotony and Training Strain Descriptions

Figure 2 shows an overall view of the weekly average for TM and TS calculated through the s-RPE across 20 weeks. Overall, Figure 2 shows that the highest TM occurred in week 12 during mid-season (9.60 AU), while the lowest value occurred in week 4 during early-season (1.92 AU). The highest TS occurred in week 17 during end-season (6656.51 AU), while the lowest value occurred in week 4 during early-season (797.17 AU).

## 4. Discussion

The aim of the present study was to quantify and analyse the relationship of the external and internal training intensity metrics as well as the well-being measures in different periods of a semi-professional soccer season (early-, mid- and end-season); and to describe TM and TS for the entire period of the analysis in a semi-professional soccer season. 

In the case of external training intensity metrics, it was observed that the total HSRD and sprint distance in comparison to the early and mid-season increased at the end-season. Previous studies had used various methods to examine these factors. According to the coaches, there could be variances in training and performance depending on the degree of play that led to differences in the values reported in different periods [26]. The number of training sessions and team competitions at the end-season was higher than the previous periods, which in turn can affect the high total HSRD and TD. The average HSRD showed a positive correlation with RPE changes. In many studies, changes in internal intensity exhibit high correlation with low speeds and low-intensity distance, but not with high speeds. Possible reasons include GPS error at high speeds, individual ability/requirement to reach high speeds, and the nonlinear relationship between speeds and internal intensity. One aspect that internal intensity does not take into account is moving at higher speeds (>14 km·h^−1^) and high accelerations (>2 m·s^−2^) [27,28]. According to previous results, RPE also showed the highest values in the end-season (4.27 AU), which shows a significant difference between the early and mid-season, and follows the same line of external intensity. 

Analysis of internal intensity measured by psychological variables such as RPE is highly preferred because of its potential for integrating different types of stimuli and ease of use [8]. Various factors may affect RPE. As an example, situations such as scoring, scoring opportunities, ball control, tackle, good play on set, winning turnover, increasing ball possession or ability to block the attack, or even non-technical/tactical training can have an impact the perceived exertion of a player [29]. Therefore, given the team’s matches and training seasons at the end-season compared to previous periods, a higher RPE value can be justified. Furthermore, s-RPE was considered an important global indicator of training intensity and intensity in team sports [29,30]. However, Haddad et al. suggested that s-RPE is not sufficient to identify health indicators such as subjective fatigue, DOMS, stress, and sleep level of young soccer players [10]. In this case, Hooper and Mackinnon [5] suggested a self-assessment-based psychometric questionnaire, which includes well-being related to sleep, stress, fatigue, and DOMS. It is considered one of the best questionnaires for estimating well-being and monitoring the perceived health of soccer players [6,19]. 

In the present study, no significant difference was observed in well-being measures between the periods of the season. Evaluating RPE was found very important instrument in correlating overtraining of athletes with physical demands on the body. However, changes in TM and TS were also not significant during the season. It is not clear why it fluctuates, exhibiting a W-shaped diagram, during the season. Several factors such as match position, match result, opponent quality, tactics system, and training program can affect these results. Contextual factors such as tactical formation and suspension of a match can affect the overall workload during a match, and further into the previous or next training session [21]. This result may have been a strategy of the coach to prepare the team for the next period to achieve better results. On the other hand, the rise in TS had led to a decline in TM in the coming weeks, which can be seen in the indicators of well-being (considering not being meaningful). The degree of sleep quality gradually decreased compared to the early-season, and the quality of sleep also improved with the increase in TS and decrease in TM in the last weeks of the end-season. Additionally, DOMS and stress were higher than the end-season. Past studies had also shown that increasing the training intensity and competitions impairs sleep quality [31,32]. On the other hand, the changes were consistent with previous literature, where the high diversity of TM and TS in the mid-season reflected that players prefer greater training intensity for motivation [33]. 

The average of TD showed a moderate to high correlation with RPE, sleep and s-RPE at early-, mid- and end-seasons. In a similar study, the daily intensity over the course of a week revealed a high and moderate correlation with peak power and change of direction at different periods of the seasons in elite youth soccer players [34]. Another study examined the daily training intensity and perceived wellness characteristics and showed that the amount of training intensity was linked to sleep perceived by elite football players. This condition was also reported in our study [6].

In the other part of results, there was a large association between TD and sprint distance and between TD and HSRD in the early-, mid- and end-season. It can be inferred that this high level of correlation indicates the effect of external training intensity indices on each other. Numerous studies have reported a negative correlation between training intensity and strength indices. However, the methods of measuring internal training intensity were different from the present study [35], which reinforces the need for more studies to confirm the present results. Additionally, a review article assessed the symptoms of perceived stress and it showed that both categories were sensitive to acute changes [36]. On the other hand, increasing HRavg is associated with increasing TM and TS, which indicates that increasing external intensity increases HRavg. On the other hand, the increase in TM leads to overtraining based on a previous study [37], which is one of the consequences of overtraining, and consequently increases the heart rate during training and competition. 

In the early-, mid-, and end-season, there was a large and negative relationship between Avg TD and DOMS, as well as between sleep and Avg HSRD. Similar to the results of this study, previous research has linked perceived sleep, stress, fatigue, and DOMS to daily perceived intensity at the professional level [38].

Meanwhile, there are limitations to this study that need to be considered. First, we can point to the lack of pre-season information, which affects the overall results. Second, the number of athletes participating in this study was relatively small, which makes it difficult to generalize the results. Third, in future studies, changes in acute and chronic training can also be considered along with external and internal training intensity to obtain complete information. Fourth, stress from HI questionnaire was not considered in the present study which could present more details on data analysis. The final limitation of this study was the lack of internal and external load monitoring in resistance training and competitions sessions which should be considered in future studies. 

## 5. Conclusions

Coaches and their staff should consider the results of this study. Despite the relationship between external and internal intensity, each metric had a unique effect on the perception of the player’s training intensity management, and special attention should be paid to each player when monitoring a training session. 

Sport coaches and fitness professionals should be wary of changes in TM and TS that affect players’ good responses, because based on the results, increasing these metrics can have a negative effect on indicators such as DOMS and sleep. 

This study shows the relationship between training intensity and other psychological indicators among players. Examining these processes will probably be effective in training planning. Therefore, in order to better control the training, more consideration should be given by the coaches, so that the team performance can be maximized, and better results can be obtained. The results serve as a useful tool for providing coaches and their staff information on determining perception factors.

## Figures and Tables

**Figure 1 biology-11-00467-f001:**
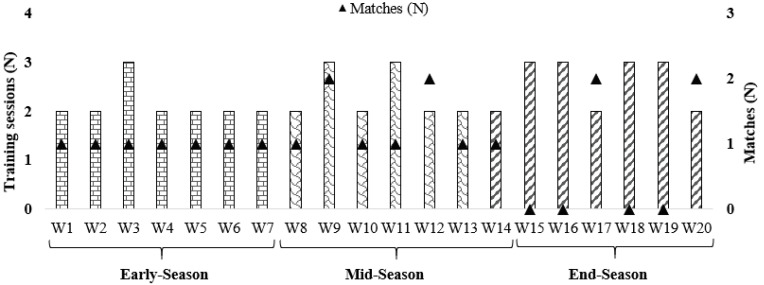
Weekly (W) distribution of training sessions and number of matches across the season.

**Figure 2 biology-11-00467-f002:**
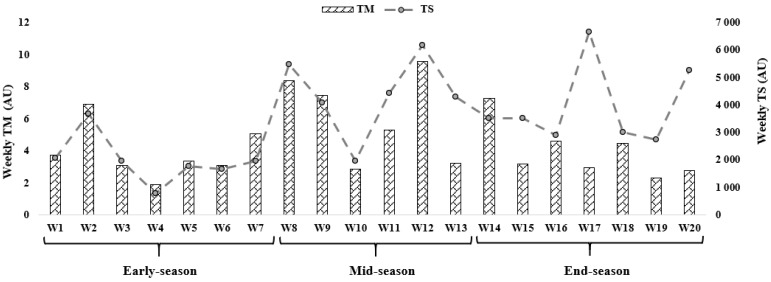
Training monotony (TM) and training strain (TS) variations calculated using the session rate of perceived exertion (s-RPE) across 20 weeks in different moments of a semi-professional soccer season.

**Table 1 biology-11-00467-t001:** Descriptive statistics (mean ± standard deviation (SD)) of the external, and internal and well-being measures in the early-, mid- and end-season.

Measure	EarS (Mean ± SD)	MidS (Mean ± SD)	EndS (Mean ± SD)	*p*	Hedges’ g (95% CI)
RPE (AU)	3.68 ± 0.62	3.65 ± 0.61	4.27 ± 0.60	EarS vs. MidS: 1.000	-
				EarS vs. EndS: 0.009	−1.75 [−2.56, −0.99] large
				MidS vs. EndS: 0.002	−1.87 [−2.70, −1.11] large
Avg TDur (Min)	58.91 ± 7.62	71.99 ± 3.72	77.44 ± 3.67	EarS vs. MidS: <0.01	−2.13 [−3.00, −1.34] very large
				EarS vs. EndS: <0.01	−3.03 [−4.07, −2.11] very large
				MidS vs. EndS: <0.01	−1.44 [−2.21, −0.73] large
Total TDur (Min)	989.61 ± 136.89	1180.22 ± 136.96	1846.77 ± 177.11	EarS vs. MidS: 0.001	−1.36 [−2.12, −0.65] large
				EarS vs. EndS: <0.01	−5.29 [−6.84, −3.96] nearly perfect
				MidS vs. EndS: <0.01	−4.11 [−5.39, −3.01] nearly perfect
s-RPE (AU)	235.89 ± 35.19	285.87 ± 26.96	333.27 ± 58.41	EarS vs. MidS: <0.01	−1.56 [−2.34, −0.83] large
				EarS vs. EndS: <0.01	−1.97 [−2.82, −1.20] large
				MidS vs. EndS: 0.028	−1.02 [−1.73, −0.34] moderate
Sleep	2.30 ± 0.75	2.13 ± 0.38	2.18 ± 0.44	EarS vs. MidS: 1.000	-
				EarS vs. EndS: 1.000	-
				MidS vs. EndS: 1.000	-
Stress	1.81 ± 0.49	1.50 ± 0.27	1.39 ± 0.29	EarS vs. MidS: 0.086	-
				EarS vs. EndS: 0.032	0.77 [0.09, 1.46] small
				MidS vs. EndS: 0.489	-
DOMS	2.18 ± 0.71	2.12 ± 0.31	2.18 ± 0.36	EarS vs. MidS: 1.000	-
				EarS vs. EndS: 1.000	-
				MidS vs. EndS: 1.000	-
TM (AU)	3.88 ± 2.44	6.14 ± 3.86	3.93 ± 0.69	EarS vs. MidS: 0.120	-
				EarS vs. EndS: 1.000	-
				MidS vs. EndS: 0.087	-
TS (AU)	1988.93 ± 1210.84	4405.95 ± 2935.36	3948.96 ± 647.78	EarS vs. MidS: 0.011	−1.05 [−1.78, −0.37] moderate
				EarS vs. EndS: <0.01	−1.97 [−2.82, −1.20] large
				MidS vs. EndS: 1.000	-
Avg TD (Km)	5.62 ± 0.86	5.24 ± 0.86	5.14 ± 0.51	EarS vs. MidS: 0.165	-
				EarS vs. EndS: 0.039	0.66 [0.001, 1.35] moderate
				MidS vs. EndS: 1.000	-
Total TD (Km)	108.30 ± 29.52	99.82 ± 0.86	133.86 ± 39.43	EarS vs. MidS: 0.493	-
				EarS vs. EndS: 0.022	−0.71 [−1.40, −0,05] moderate
				MidS vs. EndS: 0.003	−1.19 [−1.93, −0.49] moderate
Avg HSRD (Km)	0.72 ± 0.23	1.53 ± 0.29	3.07 ± 0.48	EarS vs. MidS: <0.01	−3.03 [−4.07, −2.10] very large
				EarS vs. EndS: <0.01	−6.11 [−7.85, −4.62] nearly perfect
				MidS vs. EndS: <0.01	−3.79 [−5.00, −2.75] very large
Total HSRD (Km)	14.06 ± 5.52	28.20 ± 9.47	77.95 ± 21.90	EarS vs. MidS: <0.01	−1.78 [−2.59, −1.03] large
				EarS vs. EndS: <0.01	−3.91 [−5.14, −2.84] very large
				MidS vs. EndS: <0.01	−2.88 [−3.89, −1.98] very large
Avg SD (Km)	0.61 ± 0.05	0.51 ± 0.03	0.56 ± 0.04	EarS vs. MidS: 0.079	-
				EarS vs. EndS: 1.000	-
				MidS vs. EndS: 0.638	-
Total SD (Km)	11.63 ± 4.14	9.55 ± 4.07	14.01 ± 4.00	EarS vs. MidS: 0.061	-
				EarS vs. EndS: 0.202	-
				MidS vs. EndS:0.002	−1.08 [−1.80, −0.39] moderate
HRavg (bpm)	137 ± 2	140 ± 9	135 ± 2	EarS vs. MidS: 1.000	-
				EarS vs. EndS: 1.000	-
				MidS vs. EndS: 1.000	-

Abbreviations: EarS, early-season; MidS, mid-season; EndS, end-season; AU, arbitrary units; RPE, rate of perceived exertion; Avg, average; TDur, Training duration; Min, minutes; s-RPE, session rate of perceived exertion; DOMS, delayed onset muscle soreness; TM, training monotony; TS, training strain; TD, total distance; Km, kilometres; HSRD, high-speed running distance; SD, speed distance; HRavg, heart rate average; Bpm, beats per minute.

**Table 2 biology-11-00467-t002:** Correlation analysis between the measures in study on the early-season.

Measure	β0	β1	β2	β3	β4	β5	β6	β7	β8	β9	β10	β11	β12	β13	β14	β15
RPE (β0)	1.00															
Avg TDur (β1)	**0.687 #**	1.00														
Total TDur (β2)	**0.707 §**	**0.985 £**	1.00													
s-RPE (β3)	**0.783 §**	**0.528 #**	**0.542** #	1.00												
Sleep (β4)	**0.519 #**	0.388	0.374	0.347	1.00											
Stress (β5)	−0.235	−0.120	−0.126	−0.371	0.270	1.00										
DOMS (β6)	0.338	0.263	0.234	0.364	0.465	0.051	1.00									
TM (β7)	0.047	0.109	0.129	−0.026	−0.088	0.175	0.117	1.00								
TS (β8)	0.136	0.200	0.215	0.132	0.007	0.157	0.244	**0.975 £**	1.00							
Avg TD (β9)	−0.047	−0.029	0.018	−0.080	−0.312	0.063	**−0.482 ***	−0.191	−0.267	1.00						
Total TD (β10)	−0.001	0.087	0.124	0.063	−0.248	0.103	−0.333	−0.065	−0.094	**0.930 £**	1.00					
Avg HSRD (β11)	−0.236	−0.314	−0.289	−0.295	**−0.485 ***	0.060	0.010	0.139	0.051	**0.613 #**	**0.586 #**	1.00				
Total HSRD (β12)	−0.179	−0.244	−0.215	−0.192	−0.430	0.112	−0.013	0.146	0.078	**0.701 §**	**0.733 §**	**0.970 £**	1.00			
Avg SD (β13)	0.027	0.020	−0.002	−0.052	−0.014	−0.231	0.036	−0.161	−0.164	0.425	0.339	**0.530 #**	**0.482 ***	1.00		
Total SD (β14)	0.061	0.067	0.066	0.027	−0.005	−0.097	0.032	−0.085	−0.079	**0.601 #**	**0.614 #**	**0.634 #**	**0.668 #**	**0.920 £**	1.00	
HRavg (β15)	−0.281	0.010	0.054	−0.150	−0.334	0.016	−0.331	−0.060	−0.098	0.461	0.441	0.154	0.209	0.076	0.154	1.00

Significant differences (*p* ≤ 0.05) are highlighted in bold. Abbreviations: RPE, rate of perceived exertion; Avg, average; TDur, Training duration; s-RPE, session rate of perceived exertion; DOMS, delayed onset muscle soreness; TM, training monotony; TS, training strain; TD, total distance; HSRD, high-speed running distance; SD, speed distance; HRavg, heart rate average; *, moderate effect; #, large effect; §, very large effect; £, virtually perfect effect.

**Table 3 biology-11-00467-t003:** Correlation analysis between the measures in study on the mid-season.

Measure	β0	β1	β2	β3	β4	β5	β6	β7	β8	β9	β10	β11	β12	β13	β14	β15
RPE (β0)	1.00															
Avg TDur (β1)	0.457	1.00														
Total TDur (β2)	**0.520 #**	0.439	1.00													
s-RPE (β3)	0.391	0.050	−0.005	1.00												
Sleep (β4)	**0.491 ***	0.404	**0.523 #**	−0.149	1.00											
Stress (β5)	0.241	−0.158	0.091	0.151	0.263	1.00										
DOMS (β6)	0.320	0.008	**0.499 ***	0.032	**0.518 #**	0.343	1.00									
TM (β7)	0.226	0.049	0.082	0.065	0.347	−0.229	0.022	1.00								
TS (β8)	0.285	0.039	0.093	0.234	0.300	−0.159	0.040	**0.969 £**	1.00							
Avg TD (β9)	0.320	0.444	−0.009	0.090	0.098	−0.284	−0.014	0.390	0.333	1.00						
Total TD (β10)	0.151	0.338	0.002	0.118	0.004	**−0.542 #**	−0.070	0.355	0.327	**0.852 §**	1.00					
Avg HSRD (β11)	0.263	0.104	0.080	0.122	0.096	−0.159	−0.019	−0.069	−0.128	0.403	0.378	1.00				
Total HSRD (β12)	0.184	0.270	0.082	0.255	−0.021	**−0.543 #**	−0.100	0.115	0.089	**0.646 #**	**0.839 §**	**0.725 §**	1.00			
Avg SD (β13)	0.280	0.326	−0.089	0.275	−0.148	−0.319	0.013	0.096	0.052	**0.533 #**	0.423	**0.472 ***	**0.537 #**	1.00		
Total SD (β14)	0.171	0.331	−0.038	0.309	−0.143	**−0.562 #**	−0.052	0.191	0.171	**0.671 #**	**0.825 §**	**0.507 #**	**0.878 §**	**0.790 §**	1.00	
HRavg (β15)	0.175	0.081	−0.146	0.272	0.151	−0.183	−0.194	**0.853 §**	**0.886§**	0.367	0.337	−0.020	0.132	0.189	0.239	1.00

Significant differences (*p* ≤ 0.05) are highlighted in bold. Abbreviations: RPE, rate of perceived exertion; Avg, average; TDur, Training duration; s-RPE, session rate of perceived exertion; DOMS, delayed onset muscle soreness; TM, training monotony; TS, training strain; TD, total distance; HSRD, high-speed running distance; SD, speed distance; HRavg, heart rate average; *, moderate effect; #, large effect; §, very large effect; £, virtually perfect effect.

**Table 4 biology-11-00467-t004:** Correlation analysis between the measures in study on the end-season.

Measure	β0	β1	β2	β3	β4	β5	β6	β7	β8	β9	β10	β11	β12	β13	β14	β15
RPE (β0)	1.00															
Avg TDur (β1)	0.005	1.00														
Total TDur (β2)	0.210	0.082	1.00													
s-RPE (β3)	**0.843 §**	−0.034	**0.483 ***	1.00												
Sleep (β4)	0.398	−0.264	0.168	0.298	1.00											
Stress (β5)	0.309	−0.137	0.328	**0.483 ***	0.418	1.00										
DOMS (β6)	0.342	−0.467	0.247	0.167	**0.656 #**	0.429	1.00									
TM (β7)	−0.312	0.103	0.221	−0.189	−0.402	−0.287	−0.285	1.00								
TS (β8)	0.085	0.204	**0.592 #**	0.256	−0.315	−0.152	−0.172	**0.711 §**	1.00							
Avg TD (β9)	0.404	−0.326	−0.299	0.236	0.153	−0.083	0.269	−0.361	−0.257	1.00						
Total TD (β10)	0.061	−0.361	−0.459	−0.054	−0.073	−0.244	0.029	−0.320	**−0.527 #**	**0.786 §**	1.00					
Avg HSRD (β11)	0.456	−0.346	−0.102	0.431	0.225	0.151	0.245	−0.456	−0.075	**0.706 §**	0.378	1.00				
Total HSRD (β12)	0.230	−0.363	−0.297	0.179	0.020	−0.057	0.109	**0.499 ***	−0.415	**0.841 §**	**0.867 §**	**0.751 §**	1.00			
Avg SD (β13)	0.233	−0.283	−0.080	0.144	0.284	0.097	0.223	0.038	0.187	0.073	−0.211	**0.495 ***	0.052	1.00		
Total SD (β14)	0.186	**−0.521 #**	−0.381	0.038	0.084	−0.064	0.252	−0.308	−0.339	**0.668 #**	**0.688 #**	**0.645 #**	**0.785 §**	**0.510 #**	1.00	
HRavg (β15)	0.249	0.121	−0.253	0.201	−0.066	0.169	−0.149	0.093	0.028	−0.017	−0.226	−0.104	−0.313	0.048	−0.275	1.00

Significant differences (*p* ≤ 0.05) are highlighted in bold. Abbreviations: RPE, rate of perceived exertion; Avg, average; TDur, Training duration; s-RPE, session rate of perceived exertion; DOMS, delayed onset muscle soreness; TM, training monotony; TS, training strain; TD, total distance; HSRD, high-speed running distance; SD, speed distance; HRavg, heart rate average; *, moderate effect; #, large effect; §, very large effect.

**Table 5 biology-11-00467-t005:** Correlation analysis between the measures in study on the full-season.

Measure	β0	β1	β2	β3	β4	β5	β6	β7	β8	β9	β10	β11	β12	β13	β14	β15
RPE (β0)	1.00															
Avg TDur (β1)	0.219	1.00														
Total TDur (β2)	0.355	0.454	1.00													
s-RPE (β3)	**0.772 §**	0.106	0.428	1.00												
Sleep (β4)	0.322	0.258	0.433	0.211	1.00											
Stress (β5)	−0.111	0.048	0.281	−0.008	0.426	1.00										
DOMS (β6)	0.161	0.031	0.375	0.250	0.373	0.090	1.00									
TM (β7)	0.088	−0.017	0.248	0.011	−0.373	−0.112	−0.003	1.00								
TS (β8)	0.202	0.079	0.350	0.197	−0.384	−0.157	0.073	**0.958 £**	1.00							
Avg TD (β9)	0.385	0.122	−0.127	0.086	−0.165	−0.357	−0.285	0.150	0.173	1.00						
Total TD (β10)	0.142	−0.036	−0.186	0.017	−0.326	−0.425	−0.316	0.151	0.143	**0.887 §**	1.00					
Avg HSRD (β11)	**0.496 ***	−0.312	−0.275	0.280	−0.175	−0.316	−0.048	−0.165	−0.111	**0.522 #**	0.466	1.00				
Total HSRD (β12)	0.275	−0.263	−0.291	0.181	−0.328	**−0.500 ***	−0.240	−0.069	−0.041	**0.671 #**	**0.814 §**	**0.813 §**	1.00			
Avg SD (β13)	0.390	0.059	−0.387	0.074	−0.126	**−0.528 #**	0.077	−0.116	−0.131	0.252	0.138	0.400	0.289	1.00		
Total SD (β14)	0.253	−0.022	−0.347	0.071	−0.324	**−0.640 #**	−0.062	−0.008	−0.024	**0.607 #**	**0.721 §**	**0.524 #**	**0.746 §**	**0.721 §**	1.00	
HRavg (β15)	0.154	0.149	−0.046	0.088	−0.394	−0.141	−0.189	**0.617 #**	**0.667 #**	0.405	0.279	0.088	0.084	0.030	0.098	1.00

Significant differences (*p* ≤ 0.05) are highlighted in bold. Abbreviations: RPE, rate of perceived exertion; Avg, average; TDur, Training duration; s-RPE, session rate of perceived exertion; DOMS, delayed onset muscle soreness; TM, training monotony; TS, training strain; TD, total distance; HSRD, high-speed running distance; SD, speed distance; HRavg, heart rate average; *, moderate effect; #, large effect; §, very large effect; £, virtually perfect effect.

## Data Availability

The datasets generated and analysed during the current study are available from the corresponding author on reasonable request.

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
