# Peer review of "In-Season Quantification and Relationship of External and Internal Intensity, Sleep Quality, and Psychological or Physical Stressors of Semi-Professional Soccer Players"

_biology, 2022, doi:10.3390/biology11030467_

Round 1

Reviewer 1 Report

Comments to the Author First, thanks to the authors for submitting their work. This investigation examined in-season "Quantification and Relationship of External, Internal, Sleep Quality, and Psychological or Physical Stressors of Professional Soccer Players". In general, this investigation was well-conducted, and the novel results add to the body of literature in this area. On the other hand, the work investigates an interesting topic of analysis in the field of sports training in soccer. This study offers valuable insights into the season quantification and relationship of external, internal, sleep quality, and psychological or physical stressors of professional soccer players in collegiate athletes. However, there are several changes that should be considered prior to publication. Abstract Line25: “ from Asian First League team”. I suggest changing to “) from the Asian First League team”… Line31: “ hear rate average” Did you min heart rate? Line 34: “ RPE showed a significant highest”. I suggest changing to “ RPE showed a significantly highest” Line 34: Provide more practical results for your study Line 45: Check the keywords in the mesh Introduction Line 49-51: Start with sentences that are close to the objectives of the present study. Line 54: Before stating the study hypotheses, refer to the research objectives. Line 65: reference? Line 68-71: The relationship between TM and TS with external and internal load is expressed Line 71: Instead of entering a reference in parentheses, you wrote an 8 at the end of this line. Line 72-84: It is better to add more details about the studies mentioned in this section. Line 72: Finally, this section is very short and needs to be revised due to writing problems. see studies below: Materials and methods: In the participants section, the number of training sessions and matches are mentioned, and the period of season is stated in the experimental design section. It is better to add the training session information to the second part of the method. Line 99: In some parts of the text, the sources are handwritten. Please use source writing software (such as Endnote). Line 108-109: References should be added in these sections. Line179: More details should be added in this section because there is little reference to the coach's decision. How to calculate Table 1 using GPS information and questionnaires is not stated in the method section. Result: Please add the Hedge’s g effect size and threshold's to the analysis. The results are correctly described. Discussion Line 260 – please use the term “end-season” and standardize this term in the document Line270-272: this sentence is too long, please rephrased to clarify the idea Line 272: The results of the games were not reported in this study and couldn’t be used in the discussion. Line 282: Following the previous section, the results are also mentioned here .... Conclusion The practical work applicability can be improved

Author Response

Dear Reviewer and Editor,

We thank you for your valuable comments. We have responded to all your comments point by point and made the necessary changes in the article with "track changes". We hope you get an accept the article at this stage.

Kind regards

The authors

Reviewer 1

First, thanks to the authors for submitting their work. This investigation examined in-season "Quantification and Relationship of External, Internal, Sleep Quality, and Psychological or Physical Stressors of Professional Soccer Players". In general, this investigation was well-conducted, and the novel results add to the body of literature in this area. On the other hand, the work investigates an interesting topic of analysis in the field of sports training in soccer. This study offers valuable insights into the season quantification and relationship of external, internal, sleep quality, and psychological or physical stressors of professional soccer players in collegiate athletes.

Authors: Thank you for your opinion. We follow your suggestions and all modifications are highlighted with tracked changes.

However, there are several changes that should be considered prior to publication.

Abstract Line25: “ from Asian First League team”. I suggest changing to “) from the Asian First League team”…

Authors: Done.

Line31: “ hear rate average” Did you min heart rate?

Authors: Yes, thank you. We corrected.

Line 34: “ RPE showed a significant highest”. I suggest changing to “ RPE showed a significantly highest”

Authors: Done. Thank you.

Line 34: Provide more practical results for your study

Authors: We modified the sentence accordingly.

Line 45: Check the keywords in the mesh Introduction

Authors: We removed keywords that were presented in introduction. Thank you.

Line 49-51: Start with sentences that are close to the objectives of the present study.

Authors: Dear reviewer, although we appreciate your suggestion, we followed regular guidelines for JCR journal such as Biology (MDPI) and provided our objectives in the end of the introduction section. We hope that you can understand.

Line 54: Before stating the study hypotheses, refer to the research objectives.

Authors: As mentioned in the previous comment, we opted to refer our aims in the end of introduction section.

Line 65: reference?

Authors: Line 65 presented references. Due to the changes in the manuscript, the information is now on line 83. We also corrected the reference style.

According to the TM and TS formula, which is also stated in the methodology section, TM and TS are calculated by dividing the external load of each week of exercises compared to the previous week. Therefore, these indicators are calculated from the external load

Line 71: Instead of entering a reference in parentheses, you wrote an 8 at the end of this line.

Authors: It was corrected. Thank you

Line 72-84: It is better to add more details about the studies mentioned in this section.

Authors: We changed accordingly.

Line 72: Finally, this section is very short and needs to be revised due to writing problems. see studies below.

Authors: All section was revised. Thank you

Materials and methods: In the participants section, the number of training sessions and matches are mentioned, and the period of season is stated in the experimental design section. It is better to add the training session information to the second part of the method.

Authors: We changed accordingly.

Line 99: In some parts of the text, the sources are handwritten. Please use source writing software (such as Endnote).

Authors: All references were revised. Thank you

Line 108-109: References should be added in these sections.

Authors: We have changed this section for better clarity. Thank you.

.

Authors: This section has been corrected. thank you. The stress factor was mistyped and replaced with the fatigue factor

How to calculate Table 1 using GPS information and questionnaires is not stated in the method section.

Authors: We added it accordingly.

Result: Please add the Hedge’s g effect size and threshold's to the analysis. The results are correctly described.

Authors: Hedge’s g effect size and threshold's are present in table 1, on right column.

Discussion Line 260 – please use the term “end-season” and standardize this term in the document

Authors: We revised and changed all document. Thank you

Line270-272: this sentence is too long, please rephrased to clarify the idea

Authors: The section was revised. Thank you

This section has been deleted. thank you

Line 282: Following the previous section, the results are also mentioned here .... Conclusion The practical work applicability can be improved

Authors: We tried to improve this section accordingly. Thank you

Reviewer 2 Report

The manuscript by Hobari et al. describes a set of variables, that include external and internal load, throughout the season of an Asian First League soccer team.

The authors very well describe their results, but there is a lack of meaningfulness to their findings. While I understand the difficulties of doing research in an applied setting such as professional soccer, these results would benefit from better placing the results of the current manuscript in the context of the currently available evidence.

Major points:

  • In Table 1, HRmax is very similar to HRavg, which it is either wrong (hard to believe that players kept their 'maximal' HR throughout the training sessions) or the recording of the data is not accurate/useful. Please explain.
  • Have you performed correlations with all the data across the whole season? It may strengthen some correlation and show some interesting findings. 
  • Can you discuss why internal metrics of training load (i.e., session RPE) do not reflect accurately external loads (i.e., distance, high-speed)? Can you discuss then about the utilities of these metrics based on your findings? We know from the applied and theoretical framework they're both important, but given your findings, what is new from this data that coaches/practitioners can take from this article?
  • The discussion needs a complete revision to consisently refer to the study, there are signs of copy/paste from unrelated articles that do not make sense. A good example is line 343-345, which is completely out of place. Can you explain why this is?

Minor points:

  • In the manuscript you state this is a professional soccer team, however in figure 1 it is shown that these players only train 2-3 times per week. Could you elaborate on that? Do they perform any extra training sessions? That information would be important to put in context your results.
  • You mention in the discussion that the team won/loss matches. You should provide those results in your results section if you're going to discuss them. Otherwise, please omit any information not provided as results.

Author Response

Dear Reviewer and Editor,

We thank you for your valuable comments. We have responded to all your comments point by point and made the necessary changes in the article with "track changes". We hope you get an accept the article at this stage.

Kind regards

The authors

Reviewer 2

The manuscript by Hobari et al. describes a set of variables, that include external and internal load, throughout the season of an Asian First League soccer team.

The authors very well describe their results, but there is a lack of meaningfulness to their findings. While I understand the difficulties of doing research in an applied setting such as professional soccer, these results would benefit from better placing the results of the current manuscript in the context of the currently available evidence.

Authors: Dear reviewer, thank you for your words. We address all your comments and suggestions in the manuscript with tracked changes.

Major points:

  • In Table 1, HRmax is very similar to HRavg, which it is either wrong (hard to believe that players kept their 'maximal' HR throughout the training sessions) or the recording of the data is not accurate/useful. Please explain.
    Authors: Dear reviewer, thank you for your comment. We removed HRmax.
  • Have you performed correlations with all the data across the whole season? It may strengthen some correlation and show some interesting findings. 

Authors: We added one more table regarding correlations across whole season. Thank you

  • Can you discuss why internal metrics of training load (i.e., session RPE) do not reflect accurately external loads (i.e., distance, high-speed)? Can you discuss then about the utilities of these metrics based on your findings? We know from the applied and theoretical framework they're both important, but given your findings, what is new from this data that coaches/practitioners can take from this article?
  • Authors:  Many studies have shown that HR-based criteria for internal loads are not related to HSRD. This conclusion can also be mentioned in our study. It is very important to pay attention to this point in reviewing the results. Because running at high speeds puts heavy biomechanical loads on the body. This can be partly due to the acceleration data, but there is still a connection between internal loads and external loads when we talk about HSRD. There are two main reasons for this lack of communication, including GPS error at high speeds, individual ability / need to reach high speeds, and the nonlinear relationship between speeds and internal intensity. One aspect that internal loads do not take into account is moving at higher speeds (> 14 km · h-1) and high accelerations (> 2 m · s-2). These external variables are commonly known as the most rigorous biomechanical variables, but are not considered by our internal load criteria based on our data.

The HSRD was covered between 18 and 23 km / h on a daily basis, which justifies the mismatch between internal and external loads. RPE as an indicator of internal load also depends on the level of physical fitness and mental state of the individual. The same group training can put different pressures on each player, with each player experiencing different training pressures depending on the level of physical fitness, the amount of sleep before the game or training, the individual's nutrition, and the genetic characteristics of each individual.

  • The discussion needs a complete revision to consisently refer to the study, there are signs of copy/paste from unrelated articles that do not make sense. A good example is line 343-345, which is completely out of place. Can you explain why this is?

Authors: Dear reviewer, thank you for your comment. We removed unrelated article.

Minor points:

  • In the manuscript you state this is a professional soccer team, however in figure 1 it is shown that these players only train 2-3 times per week. Could you elaborate on that? Do they perform any extra training sessions? That information would be important to put in context your results.

Authors: Dear reviewer, thank you for your comment. Due to the lack of some information about internal and external load monitoring sessions such as resistance training and competitions sessions, we had a small number of sessions in some weeks in this study, which we presented in the restrictions section.

You mention in the discussion that the team won/loss matches. You should provide those results in your results section if you're going to discuss them. Otherwise, please omit any information not provided as results.

Authors: Dear reviewer, thank you for your comment. We removed the discussion of the team won/loss matches.

Reviewer 3 Report

This study is interesting but it is difficult to understand at parts of the manuscript. The writing is very poor with a large number of grammatical errors. Writing needs to be consistent when using past tense. The methodology is adequate but much improvement is needed from the writing side. Please see below.

Lines 2-4: The title needs to be revised. Internal and external? What do you mean?

Line 20: Do you mean “variations of external and internal well-being measures”?

Lines 22-23: Internal and external what? Stressors?

Line 25: What is the “entire period of the analysis”? Please define.

Line 30: Rate of perceived exertion

Line 31: Heart rate average

Line 32: Sprint distance

Line 42: What is meant by “good responses”? Please define.

Line 49: “..training programs on soccer players…”

Line 65: “…allow the identification of intensity variations..”

Line 74: “..the between Hooper Index (HI)..”

Line 82: “…needed to specially analyse…” – specially analyse? What is meant here?

Lines 84-87: For aim a) and b) these are essentially the same. A) is a description and B) is an analysis.

Please revise.

Line 109: “..cool-down phases…”

Line 118: “Australia”

Line 137: “Australia”

Line 149: “..for use until the end..”

Line 191: p>0.05

Lines 191-192: “…a repeated measures ANOVA test with a Bonferroni post-hoc test was used to compare variables for periods throughout the in-season.”

Line 193: “Statistical significance was set at p ≤ 0.05.”

Lines 195-197: Need to revise the symbols for ES ranges e.g. 0.2 to ≤0.6. Delete the < after the first ESs every range provided.

Line 200: Delete “The” prior to “Table 1”

Lines 200-204: Most of these abbreviations have been defined earlier. There is no need to repeat.

Line 245: Delete “The” prior to “Figure 2”

Line 268: What is meant by “special ventilation”?

Line 272: “winning period”

Line 298: “….technical staff had..”

Lines 298-304: Please be consistent with writing in the past tense. This goes for throughout all the manuscript.

Line 316: “close relationship”?

Line 322: Need to revise this sentence due to being grammatically incorrect.

Lines 326-327: “…which makes it difficult to generalize the results.”

Line 328: But you mentioned the study was likely to be underpowered due to small sample size?

Lines 335-337: Very poorly written. Please revise.

Line 344: “mental factors”?

Author Response

Dear Reviewer and Editor,

We thank you for your valuable comments. We have responded to all your comments point by point and made the necessary changes in the article with "track changes". We hope you get an accept the article at this stage.

Kind regards

The authors

Reviewer 3

This study is interesting but it is difficult to understand at parts of the manuscript. The writing is very poor with a large number of grammatical errors. Writing needs to be consistent when using past tense. The methodology is adequate but much improvement is needed from the writing side. Please see below.

Authors: Dear reviewer, thank you for your feedback. We provide a full revision of English grammar/spelling in order to avoid writing errors. All your comments and suggestions were highlighted in the manuscript with tracked changes.

Lines 2-4: The title needs to be revised. Internal and external? What do you mean?

Authors: Thank you for attention. It should be internal and external intensity. Thus, we modified the title to: “In-season Quantification and Relationship of External, and Internal Intensity, Sleep Quality, and Psychological or Physical Stressors of Professional Soccer Players”

Line 20: Do you mean “variations of external and internal well-being measures”?

Authors: We correct it to: “The purpose of this study was two-fold: a) to describe and analyse the relationship of the in-season variations of external and internal intensity metrics as well as well-being measures…”

Lines 22-23: Internal and external what? Stressors?

Authors: It should “intensity”. We add it accordingly.

Line 25: What is the “entire period of the analysis”? Please define.

Authors: We replace it by “20 weeks”.

Line 30: Rate of perceived exertion

Authors: Done

Line 31: Heart rate average

Authors: Done

Line 32: Sprint distance

Authors: Done

Line 42: What is meant by “good responses”? Please define.

Authors: We replace it by “performance”.

Line 49: “..training programs on soccer players…”

Authors: Done

Line 65: “…allow the identification of intensity variations..”

Authors: Done

Line 74: “..the between Hooper Index (HI)..”

Authors: Done

Line 82: “…needed to specially analyse…” – specially analyse? What is meant here?

Authors: We removed the word “specially”. Thank you

Lines 84-87: For aim a) and b) these are essentially the same. A) is a description and B) is an analysis.

Please revise.

Authors: We revised in both introduction and abstract sections. Thank you

Line 109: “..cool-down phases…”

Authors: Done

Line 118: “Australia”

Authors: Done

Line 137: “Australia”

Authors: Done

Line 149: “..for use until the end..”

Authors: Done

Line 191: p>0.05

Authors: Done

Lines 191-192: “…a repeated measures ANOVA test with a Bonferroni post-hoc test was used to compare variables for periods throughout the in-season.”

Authors: Done

Line 193: “Statistical significance was set at p ≤ 0.05.”

Authors: Done

Lines 195-197: Need to revise the symbols for ES ranges e.g. 0.2 to ≤0.6. Delete the < after the first ESs every range provided.

Authors: Done

Line 200: Delete “The” prior to “Table 1”

Authors: Done

Lines 200-204: Most of these abbreviations have been defined earlier. There is no need to repeat.

Authors: We change the sentence to avoid unnecessary repetitions.

Line 245: Delete “The” prior to “Figure 2”

Authors: Done

Line 268: What is meant by “special ventilation”?

Authors: According to the reviewer’s idea, the sections that discussed the outcome of the matches were replaced with related topics.

Line 272: “winning period”

Authors: Done

Line 298: “….technical staff had..”

Authors: Done

Lines 298-304: Please be consistent with writing in the past tense. This goes for throughout all the manuscript.

Authors: We conducted a revision in the entire manuscript to improve the writing. Thank you

Line 316: “close relationship”?

Authors: We changed to “large association”

Line 322: Need to revise this sentence due to being grammatically incorrect.

Authors: We review it accordingly.

Lines 326-327: “…which makes it difficult to generalize the results.”

Authors: Done

Line 328: But you mentioned the study was likely to be underpowered due to small sample size?

Authors: We removed the sentence accordingly.

Lines 335-337: Very poorly written. Please revise.

Authors: thank you, we rewrite it.

Line 344: “mental factors”?

Authors: Yes, thank you. We correct it.

Round 2

Reviewer 2 Report

I would like to thank the authors for addressing some of the points and improving the manuscript. However, I believe they have not provided enough information as to why some data/writing were not appropriately formatted/provided in the first version.

Previous points that need further explanation:

  • In Table 1, why was HRmax similar to HRavg? Please explain.
  • The discussion had signs of copy/paste from unrelated articles in line 343-345, which was completely out of place. Can you explain why this is?

Minor points:

  • Can you describe how you have calculated Effect Sizes given the magnitude of some of those?

Author Response

Comments and Suggestions for Authors

I would like to thank the authors for addressing some of the points and improving the manuscript. However, I believe they have not provided enough information as to why some data/writing were not appropriately formatted/provided in the first version.

Authors: Thank you for your comments.

Previous points that need further explanation:

  • In Table 1, why was HRmax similar to HRavg? Please explain.

Authors: Dear reviewer, as we mentioned in previous revision, we opted to remove all data regarding HRmax since there were some mistake in our data set. The HRavg is correct. Thank you so much for your consideration.

  • The discussion had signs of copy/paste from unrelated articles in line 343-345, which was completely out of place. Can you explain why this is?

Authors: We tried to provide a possible explanation for the higher values found in the end-season and justify with the contextual variable of match location. However, we did not address this variable in this study, and we agree with the reviewer that our justification is too speculative. Therefore, we removed the sentences accordingly. Thank you for you attention.   

Minor points:

  • Can you describe how you have calculated Effect Sizes given the magnitude of some of those?

Authors: We add the information on the statistical analysis accordingly. Thank you

Reviewer 3 Report

Well done on addressing my comments.

Author Response

Thank you so much for your valuable comments.

Round 3

Reviewer 2 Report

The authors have addressed all my comments.